# Defense Mechanisms of Cotton *Fusarium* and *Verticillium* Wilt and Comparison of Pathogenic Response in Cotton and Humans

**DOI:** 10.3390/ijms232012217

**Published:** 2022-10-13

**Authors:** Mingwu Man, Yaqian Zhu, Lulu Liu, Lei Luo, Xinpei Han, Lu Qiu, Fuguang Li, Maozhi Ren, Yadi Xing

**Affiliations:** 1Zhengzhou Research Base, State Key Laboratory of Cotton Biology, School of Agricultural Sciences, Zhengzhou University, Zhengzhou 450001, China; 2State Key Laboratory of Cotton Biology, Institute of Cotton Research, Chinese Academy of Agricultural Sciences, Anyang 455000, China; 3School of Pharmaceutical Sciences (Shenzhen), Shenzhen Campus of Sun Yat-Sen University, Shenzhen 518107, China; 4Hainan Yazhou Bay Seed Laboratory, Sanya 572000, China; 5Institute of Urban Agriculture, Chinese Academy of Agricultural Sciences, Chengdu 610000, China

**Keywords:** cotton, disease resistance genes, *Fusarium* wilt, *Verticillium* wilt, fungal keratitis, neutrophil

## Abstract

Cotton is an important economic crop. *Fusarium* and *Verticillium* are the primary pathogenic fungi that threaten both the quality and sustainable production of cotton. As an opportunistic pathogen, *Fusarium* causes various human diseases, including fungal keratitis, which is the most common. Therefore, there is an urgent need to study and clarify the resistance mechanisms of cotton and humans toward *Fusarium* in order to mitigate, or eliminate, its harm. Herein, we first discuss the resistance and susceptibility mechanisms of cotton to *Fusarium* and *Verticillium* wilt and classify associated genes based on their functions. We then outline the characteristics and pathogenicity of *Fusarium* and describe the multiple roles of human neutrophils in limiting hyphal growth. Finally, we comprehensively compare the similarities and differences between animal and plant resistance to *Fusarium* and put forward new insights into novel strategies for cotton disease resistance breeding and treatment of *Fusarium* infection in humans.

## 1. Introduction

Cotton (*Gossypium* spp.) is an important cash crop as it yields grain, fiber, and oil [1]. Cotton is the world’s most important source of natural fiber, providing approximately 35% of the world’s total fiber [2]. Compared with synthetic fiber, cotton is a renewable resource and cotton plantations generate a higher number of jobs that involve planting, processing, and textile manufacturing. Cotton has important environmental and social benefits, and cottonseed can be used as animal feed after oil extraction [3]. Cotton is grown in more than 80 countries, of which approximately 30 regard cotton as a major crop [4]. China is the world’s largest producer of cotton fiber [5]. However, cotton has been affected by various pests and pathogens, among which *Fusarium* and *Verticillium* wilt cause serious global cotton economic losses [6,7], thus leading to them being considered as the main obstacle to sustainable high-quality cotton production in China [8]. Therefore, this review describes the relationship between cotton and pathogenic fungi, using *Fusarium* and *Verticillium* as examples.

*Fusarium oxysporum* (*FOV*) is the cause of *Fusarium* cotton wilt and is classified into eight physiological types, of which three exist in China [9]. Based on their symptoms, there are two species of *Verticillium dahliae* that infect cotton, namely defoliating and non-defoliating strains [10]. These pathogens can exist in soil and plant debris in the form of mycelium, chlamydia spores, and microsclerotia and survive in soil for a long time (until the beginning of a new infection cycle) [9]. In addition, they can survive saprophytically on other crops and weeds [8]. Diseases caused by *Fusarium* typically manifest following the seedling stage and are most severe during the budding stage. The onset of *Fusarium* wilt is characterized by plaques between the main veins, while the rest of the leaves remain green; chlorotic plaques gradually cause the leaves to become necrotic and fall off the stem [9]. *Verticillium* wilt occurs before the budding stage, manifesting as yellow mottled or fallen leaves during the boll setting stage, when the disease is also the most serious [10]. However, the determinants that lead to the development of these diseases remain unclear [8].

Humans are warm-blooded, their innate and adaptive immune system is complex, and they are naturally resistant to most invasive fungal infections. For fungi to successfully infect humans they must: (1) grow at or above human body temperature (37 °C), (2) bypass or penetrate host surface barriers to reach internal tissues, (3) digest and absorb human tissues, and (4) resist against the human immune system. Thus, the number of fungi capable of infecting a healthy human host is very limited [11,12]. However, the pathogenic fungi *Fusarium* and *Aspergillus* have these abilities and can therefore infect humans. Although there are limited data on the role of host defenses against these fungi, *fusariosis*, *aspergillosis*, and other fungal infections share many features [13]. Hence, this review discusses *Fusarium* to illustrate the relationship between pathogenic fungi and humans. *Fusarium* can produce mycotoxins that pose a significant health risk to humans and animals. There has been a gradual increase in infections caused by *Fusarium* in immunosuppressed patients that have undergone organ transplantation and chemotherapy in recent years [14]. Moreover, *Fusarium* is second only to *Candida albicans* and *Aspergillus* in terms of morbidity and mortality in immunodeficient individuals and is internationally classified as an important infectious disease [15].

To eliminate, or significantly mitigate, the harm of *Fusarium* and *Verticillium* wilt on cotton so as to improve yield, and simultaneously seek a method to treat patients suffering from *Fusarium* spp. infectious diseases, we review the studies on cotton genes related to resistance and susceptibility to *Fusarium* and *Verticillium* wilt in recent years, summarize the molecular mechanisms by which humans resist *Fusarium* disease, compare resistance strategies in humans and plants, and provide novel insights into the improvements in the resistance of plants and humans against *Fusarium*.

## 2. Molecular Mechanisms of Plant Resistance to *Fusarium* and *Verticillium* Wilt

Plants can use various strategies to counter invasion by *Fusarium* and *Verticillium* by activating complex cellular and molecular regulatory networks in vivo. This includes using pattern recognition receptors (PRRs) to induce immune responses, modifying cell walls to block pathogens, producing extracellular enzymes to directly degrade pathogens, or activating the expression of specific genes through various signaling pathways and transcription factors [16].

In recent years, considerable research has been carried out on the physiological and molecular mechanisms of cotton *Fusarium* wilt caused by *FOV* and *Verticillium* wilt caused by *V. dahliae* [2]. This review highlights the relatively recent discovery of genes involved in defenses against *Fusarium* and *Verticillium* and classifies them according to whether they act in signaling cascades and transcriptional regulation or are directly involved in cell wall and protein defense (Figure 1). This provides a theoretical basis to better understand the molecular genetic mechanisms of plant resistance to *Fusarium* and *Verticillium* wilt and provides a reference for future studies on genetic resistance mechanisms to fungal wilt.

### 2.1. Genes Involved in Signaling Cascades and Transcriptional Regulation

During the long-term evolution of plants, an immune system directly controlled by a series of cross-linked signal transduction pathways and various plant hormones has been developed to rapidly regulate gene expression, thereby adapting, resisting, and tolerating various biotic stresses. The reaction substrates of these intricate signaling networks are usually transcription factors (TFs). TFs can be regulated via multiple signals and can also regulate the expression of multiple genes related to disease resistance. They also play an important regulatory role in the process of plant stress signal transmission.

Within this intricate signaling network, the mitogen-activated protein kinase (MAPK) cascade response is the primary pathway for sorting and amplifying external signals into intracellular signals, including three kinases—MAPK kinase kinases (MAPKKKs), MAPK kinases (MKKs), and MAPKs—which are important in response to biotic and abiotic stresses [17,18]. Infection of cotton by *FOV* significantly induced *GhMPK20* expression, whereas silencing *GhMPK20* enhanced resistance to *Fusarium* cotton wilt. The WRKY family of TFs regulate various biotic stress responses and physiological processes. GhWRKY40 is a substrate of GhMPK20; *FOV* induces *GhWRKY40* expression through the GhMKK4-GhMPK20 cascade reaction, whereas silencing of *GhMKK4* and *GhWRKY40* enhances resistance to *Fusarium* cotton wilt [19]. However, *GhWRKY53* silencing activates the jasmonic acid (JA) signaling pathway and inhibits the salicylic acid (SA) signaling pathway, which reduces resistance to *Verticillium* cotton wilt [20]. Other members of the WRKY family are regulated by other hormones. *GbWRKY1* is a negative regulator of the JA-mediated defense pathway involved in plant resistance to *V. dahliae* and mediates the transition from defense to development by activating JAZ1 expression during infection by *V. dahliae* [21]. *GhWRKY48* expression in roots, stems, and leaves is induced by *V. dahliae* infestation, JA, and SA; whereas its silencing enhances resistance to *Verticillium* cotton wilt and may act as a negative regulatory transcription factor to repress the expression of downstream disease resistance genes, thereby affecting resistance to *Verticillium* cotton wilt [22]. MKK family members play a dual role in regulating resistance to fungal pathogens in cotton; *GhMKK4*, *GhMKK6*, and *GhMKK9* positively regulate resistance to *Verticillium* wilt, whereas *GhMKK10* negatively regulates resistance to *Verticillium* wilt [23]. MAPK scaffold protein (GhMORG1) interacts with GhMKK6 and GhMPK4, and its overexpression significantly enhances the activity of the GhMKK6-GhMPK4 cascade response and positively regulates resistance to *Fusarium* cotton wilt [24]. GhAP2C1 is a negative regulator of the MAPK cascade that interacts with GhMPK4 to negatively regulate the immune response to *Fusarium* cotton wilt [25]. Gene silencing experiments demonstrated that *GhNDR1* and *GhMKK2* are essential for cotton resistance to *Verticillium* wilt [26].

Glutathione transferases (GSTs) catalyze their binding to glutathione for various physiological functions, including disarming endogenous and exogenous toxins, participating in the translocation of flavonoids, and affecting disease resistance in plants [27]. *GaGSTF9* overexpression positively regulates resistance to *Verticillium* wilt in *Arabidopsis* [28]. Similarly, *GbGSTU7* positively regulates resistance to *Fusarium* wilt in *Gossypium barbadense,* whereas its silencing significantly reduces glutathione peroxidase activity in vivo and increases the incidence of *Fusarium* cotton wilt [29].

Expression of ribosomal protein GaRPL18 is induced by SA treatment, suggesting that *GaRPL18* is associated with SA-mediated signaling pathways. Silencing *GaRPL18* significantly reduces the number of immune molecules induced by the SA signaling pathway and reduces *Verticillium* wilt resistance in cotton, whereas *GaRPL18* overexpression increases plant resistance to *Verticillium* wilt [30]. Expression of cotton cyclin-dependent kinase E (*GhCDKE*) is induced by *Verticillium* wilt infestation and methyl jasmonate (MeJA); *GhCDKE* silencing in cotton increases vulnerability to *Verticillium* wilt, whereas its overexpression in *Arabidopsis* enhances resistance [31]. The JA-induced calcium-dependent protein kinase *GhCPK33* negatively regulates resistance to *Verticillium* cotton wilt, and knockdown of *GhCPK33* enhances resistance [32]. The expression of *GhIQM1* Ca^2+^-independent calmodulin-binding protein is induced by *V. dahliae* and SA, which may negatively regulate resistance to *Verticillium* cotton wilt by inhibiting the SA pathway [33]. *GhBIN2* interacts with JAZ (jasmonate-ZIM domain) family proteins and phosphorylates JAZ proteins in response to infestation by *V. dahliae*. This may negatively regulate resistance to *Verticillium* cotton wilt by regulating the JA signaling pathway, since *GhBIN2* overexpression in cotton increases sensitivity to *Verticillium* wilt [34].

Several stress factors induce the production of large amounts of reactive oxygen species (ROS) in plants, which act as direct toxic agents against pathogens, and may act as signaling molecules for defense; ROS levels are precisely regulated by the plant [35,36]. Ascorbic acid (AsA) is an important cellular antioxidant that oxidizes ROS and affects signaling pathways involving ROS, while oxidized inositol can be involved in AsA synthesis [37]. Inhibition of *myo*-inositol oxygenase *GbMIOX5* causes H_2_O_2_ accumulation, and a reduction in AsA content in cotton cells significantly reduces resistance to *Verticillium* cotton wilt [38].

Lipids are essential plant cell components and can act as signaling molecules to regulate plant pathogen resistance [39]. Silencing of stearoyl-ACP desaturase *GhSSI2* enhances resistance to *Fusarium* and *Verticillium* cotton wilt. Inhibition of *GhSSI2* expression reduces oleic acid content, which adversely affects cell membrane function and activates the immune response, leading to increased SA content. SA triggers a hypersensitive response (HR) in plants, with rapid accumulation of ROS and induction of high levels of PR (pathogenesis-related) gene expression, which seals the pathogen in the tomb of dead cells [40]. Oleic acid is also a defense signal which, at low levels, can directly upregulate the expression of different disease resistance genes in a pathway independent of SA and JA, whereas activating the second messenger, NO (nitric oxide), upregulates a wide range of intranuclear disease resistance genes [41].

Knockout of the SA-related spermine (Spm) protein, Spm synthase (GhSPMS), and S-adenosylmethionine decarboxylase (*GhSAMDC*) genes increases the susceptibility to *Verticillium* cotton wilt. Conversely, transfection of *Arabidopsis* with these genes enhances its resistance to *Verticillium* wilt, indicating that the SA signaling pathway and Spm biosynthesis are associated with plant resistance to *Verticillium* wilt [42]. Expression of cotton polyamine oxidase (*GhPAO*) in *Arabidopsis* increases its resistance to *Verticillium* wilt and causes significant accumulation of H_2_O_2_, SA, and camalexin (a plant antitoxin), implying that GhPAO promotes plant resistance to *Verticillium* wilt by activating signaling pathways associated with Spm and camalexin [43]. 

The MYB (v-myb avian myeloblastosis viral oncogene homolog) transcription factor family is involved in resistance to pathogen infection, and knockout of *GhMYB108* makes cotton more susceptible to *Verticillium* wilt, whereas *GhMYB108* overexpression improves resistance to *Verticillium* wilt in *Arabidopsis* [44]. Silencing the homeodomain transcription factor (*HDTF1*) activates the JA signaling pathway and further improves *Verticillium* cotton wilt resistance [45]. Both JA and *V. dahliae* can induce the expression of HD-ZIP transcription factor *GhHB12*, with its overexpression reducing the resistance of cotton to *V. dahliae* [46]. *GbbHLH171* interacts with, and is phosphorylated by, a defense-associated receptor-like kinase (*GbSOBIR1*) and positively regulates resistance to *Verticillium* cotton wilt [47].

Ethylene (ET) response factors (ERFs) are required for pathogen defense responses, and genes from different hormone signaling pathways all play important roles in response to infection by fungal pathogens. *GbABR1* of the apetala 2 (AP2) family positively regulates plant resistance to *Verticillium* wilt, whereas its silencing increases disease rates in cotton [48]. *GbERFb* is an AP2/ERF type TF which increases disease resistance in cotton [49]. *GbERF1*-like ERFs enhance plant resistance to *Verticillium* wilt by positively regulating lignin synthesis [50]. *V. dahliae* and phytohormones (SA, JA, and ET) induce the expression of the nucleotide binding site–leucine-rich repeat (NBS-LRR) gene *GbaNA1* in cotton, which is involved in its response to *Verticillium* wilt [51]; *GbaNA1* overexpression increases ROS levels in *Arabidopsis* and promotes the expression of genes related to the ET signaling pathway [52]. The CC-NBS-LRR gene *GbCNL130* activates the SA-mediated signaling pathway, increases ROS accumulation, and promotes the expression of downstream PR genes, thereby enhancing resistance to *Verticillium* cotton wilt [53]. In addition, knockdown of positively regulated genes *GhJAZ10*, *GhbHLH18* (JA), *GhPUB17* (SA), and *GhEBF1* (ET) significantly increases susceptibility to *Verticillium* wilt in resistant varieties, whereas silencing of negatively regulated genes *GhTGA7* (SA) and *GhBZR1* (Brassinolide, BR) significantly increases resistance to *Verticillium* wilt in susceptible varieties [54].

The bZIP transcription factor *GbVIP1* (VirE2 interacting protein 1) is induced by *V. dahliae* and ET, and its silencing reduces *Verticillium* cotton wilt resistance, whereas its ectopic expression in tobacco enhances resistance to *Verticillium* wilt in tobacco by positively regulating the expression of *PR1*, *PR1*-like, and *HSP70* genes [55]. JA induces the expression of *BEL1*-like transcription factor *GhBLH7-D06* in vascular tissues, acting on secondary cell formation and negatively regulating resistance to *Verticillium* cotton wilt [56]. In addition, knockdown of transcription factor *GbNAC1* reduces resistance to *Verticillium* cotton wilt, whereas its overexpression increases *Verticillium* wilt resistance in *Arabidopsis* [57].

### 2.2. Genes Directly Involved in Cell Wall and Protein Defense

Plant resistance to fungal pathogens is largely influenced by defense-related proteins. Plants have evolved sophisticated sensory mechanisms to detect microbial invasion and overcome the negative effects caused by microbes in terms of growth, yield, and survival [58].

*V. dahliae* invades host plants mainly through plant cell wall degradation; therefore, the cell wall is the first barrier for plants that helps prevent pathogen entry, often effectively stopping pathogen invasion [59,60]. Pectin is one of the main components of the plant cell wall, and its structure and function are closely related to the degree of esterification. Pectin methylesterases (PMEs) protect plant cell walls by catalyzing the dimethyl esterification of pectin polygalacturonate. *GhPMEI3* regulates resistance to *V. dahliae* in cotton since its silencing increases susceptibility to *V. dahliae* infestation, while ectopic expression of *GhPMEI3* increases pectin methyl esterification and limits fungal infection by regulating root growth [61]. Knockout of pectin lyase genes *VdPL3.1* and *VdPL3.3* reduces pathogen virulence to cotton [60]. Cell wall polysaccharides are mainly modified by acetylation; the level and location of this modification directly affects the structure, physicochemical properties, and ability of the cell wall to resist pathogens [62]. Acetyltransferase *GhTBL34* (trichome birefringence-like) expression is induced by *V. dahliae*, ET, SA, and JA and mediates cell wall polysaccharide acetylation to block pathogen invasion and regulate *Verticillium* wilt resistance [63,64]. Resistance of cotton to wild-type fungal diseases is largely dependent on lignin content, and its levels can be used as an indicator of resistance against *Fusarium* wilt [65]. Transcriptome analysis suggests that caffeic acid 3-O-methyltransferase (COMT) and peroxidase 2 (POD2) may be involved in the synthesis and accumulation of lignin in response to *Fusarium* cotton wilt [66]. Patatin-like proteins (PLPs) are defense proteins with non-specific lipid acyl hydrolytic activity that hydrolyze cell membranes into fatty acids and lysophospholipids. *GhPLP2,* which is induced by *FOV*, *V. dahliae*, ET, and JA, plays an important role in reducing fungal pathogenicity and regulating plant resistance to *Verticillium* wilt [67]. Overexpression of polygalacturonase inhibitor proteins *CkPGIP1* and *GhPGIP1* from *Cynanchum komarovii* and *G. hirsutum* enhances cotton resistance to *Verticillium* wilt by promoting xylem lignification in cotton [68]. The apoplastic thioredoxin protein *GbNRX1* enhances the immune response of cotton to *V. dahliae*; *GbNRX1* silencing causes a large accumulation of ROS and increases susceptibility to *V. dahliae* infection [69].

BLADE-ON-PETIOLE (BOP) 1 and BOP2 are two BTB-ankyrin proteins specifically expressed at lateral-organ boundaries (LOBs). *GhBOP1* acts synergistically with *GhBP1* to co-regulate lignin biosynthesis and enhance resistance to *Verticillium* cotton wilt [70]. *GbHyPRP1* encodes a proline-rich protein with a POLE1 structural domain that negatively regulates resistance to *Verticillium* cotton wilt. Expression of *GbHyPRP1* is inhibited by the SA signaling pathway when cotton is infected with *V. dahliae*, and *GbHyPRP1* silencing enhances resistance to *V. dahliae* by thickening the cell wall and accumulating ROS; in contrast, *HyPRP1* overexpression reduces resistance to *Verticillium* wilt in *Arabidopsis* [71].

Latex protein *GhMLP28* is induced by *V. dahliae*, JA, SA, or ET to produce and regulate plant disease resistance [2]. *GhDIR1* overexpression increases cotton lignin content and enhances resistance to *Verticillium* cotton wilt [72]. Defense regulator EDS1 encodes a lipase-like protein that is induced by SA and regulates plant resistance to *Verticillium* wilt. *GbEDS1* overexpression promotes the production of SA and H_2_O_2_ and enhances *Arabidopsis* resistance to *Verticillium* wilt, whereas *GbEDS1* silencing reduces cotton resistance to *Verticillium* wilt [73].

Papain-like cysteine proteases (PLCPs) play an important role in plant defense against pathogen invasion. *GhRD21-7* substantially promotes the resistance to *Verticillium* cotton wilt and its overexpression further enhances this resistance [74]. Germin-like proteins (GLPs) are a family of glycoproteins involved in plant resistance to various stresses. *GhGLP2* silencing increases susceptibility to *FOV* and *V. dahliae* infection in cotton, whereas *GhGLP2* overexpression attenuates mycelial growth of these fungi, increases callus formation at leaf infection sites, enhances cell wall lignification, and improves resistance to *Fusarium* and *Verticillium* wilt in *Arabidopsis* [75]. Thaumatin-like proteins (TLPs) are a family of multi-member defense-related proteins; *GhTLP19* silencing increases malondialdehyde levels and decreases catalase levels in cotton, which increases its susceptibility to *Verticillium* wilt [76].

Extracellular enzymes help plants to resist fungal pathogen infection. For example, chitinase (Chi) hydrolyzes chitin (a main component of the fungal cell wall) into N-ethoxylated oligosaccharides and glucose to inhibit spore germination and mycelial growth [77]. There are low levels of Chi (with low activity) in all higher plant organs; however, its levels rapidly increase following pathogen infestation [78]. *Chi23*, *Chi32*, or *Chi47* positively regulate *Verticillium* wilt resistance; their knockout reduces resistance to *Verticillium* cotton wilt [77]. Cotton infestation by *V. dahliae* triggers the secretion of Chi28 and cysteine protein CRR1 in order to degrade the fungal cell wall and protect Chi28 from degradation by serine protease secreted by *V. dahlia,* respectively. Thus, silencing either *Chi28* or *CRR1* reduces resistance to *Verticillium* cotton wilt, whereas *CRR1* overexpression enhances its resistance [79]. Plant lysine motif-containing (LysM) proteins recognize chitin in the cell wall of fungal pathogens and initiate the corresponding defense response. Silencing *Lyp1*, *Lyk7,* and *LysMe3* significantly reduces the production of SA, JA, and ROS, decreases the activity of defense-related genes, and ultimately reduces resistance to *Verticillium* cotton wilt [80].

*Verticillium* highly induces laccase *GhLAC15* expression, and the cell walls of overexpressing plants enhance resistance to *Verticillium* wilt through a defense-mediated response that enhances lignification and increases arabinose and xylose accumulation [81]. GhUMC1, a divalent copper-binding protein, is an umecyanin-like gene involved in resistance to *Verticillium* cotton wilt through regulation of the JA signaling pathway and lignin metabolism [82]. Walls are thin (*WAT*) genes regulate SA metabolism and signaling by affecting polar transport of growth hormones to further enhance plant resistance to a variety of pathogens [83]. Knockout of *GhWATs* increases SA accumulation, activates the expression of SA pathway-related genes, and increases lignin accumulation in the xylem, each of which promotes plant resistance to *Verticillium* wilt [84]. Simultaneous silencing of *GhWAT1, GhWAT2*, and *GhWAT3* increases lignin accumulation in the xylem and enhances plant resistance to *Verticillium* wilt but inhibits plant growth [84]. Knockdown of the *Gh4CL30* lignin synthesis gene in cotton decreases flavonoid, lignin, and butyryl content but increases guaiacyl, caffeic acid, and ferulic acid content, which provides a new idea for resistance to *Verticillium* cotton wilt [85].

Enoyl-CoA reductase (GhECR) is involved in the formation of long-chain fatty acids, and its silencing increases the susceptibility of cotton to *FOV* and *V. dahliae* infection [86]. The subtilase-like protein GbSBT1 is highly induced following infection with *V. dahliae*, or by JA and ET treatment. The protein acts at the cell membrane and regulates plant resistance to *Fusarium* and *Verticillium* wilt [87].

Plants infested with pathogens over a long period of time have evolved various innate immune mechanisms [88]. In general, plants employ two sets of defense strategies: microbial (or pathogen)-associated molecular pattern (MAMP/PAMP)-triggered immunity (MTI/PTI) and effector-triggered immunity (ETI) [89]. Some pathogenic bacteria disrupt plant tissue structure and produce damage-associated molecular patterns (DAMPs) such as cell wall fragments, which induce innate immunity similar to that of MAMPs [90].

Activation of plant innate immunity requires receptor recognition of pathogens. MTI/PTI is triggered by PRRs located across the cell surface, whereas ETI is caused by NBS-LRR receptor proteins located intracellularly [91]. PRRs include receptor-like proteins (RLPs) and receptor-like kinases (RLKs) that are located on plant cell membranes and which are responsible for detecting molecular markers of invading pathogens or damaged plant tissues and activating defense responses, thereby preventing infection before pathogen invasion of the plant cell [92,93]. The *GhRLPGSO1*-like gene, *GhRLP44*, *GhRLP6*, and *GhRLP34* may play a role in resistance to *Fusarium* cotton wilt [94]. *Verticillium* wilt infestation of disease-resistant *G. barbadense* cotton induces *GbRLK*; transfection of cotton and *Arabidopsis* with this gene increases resistance to *Verticillium* wilt [95]. The TIR-NBS-LRR protein encoded by *CG02* is likely to be a key gene for resistance to *Verticillium* cotton wilt since it specifically promotes the expression of disease resistance genes, while silencing this gene increases its susceptibility to *Verticillium* wilt [96].

Invasion of cotton cells by *FOV* causes the release of large amounts of intracellular Ca^2+^, which acts as a second messenger to initiate the expression of various downstream disease resistance genes [97]. *GhGLR4.8* encoding glutamate-like receptor (GLR) protein plays an active role in resistance to *Fusarium* cotton wilt and it is hypothesized that it may be the ion channel required for Ca^2+^ to flood into the cell and enhance the disease resistance response [98].

In addition, the *Ve* R-gene enhances plant resistance to *Verticillium* wilt by encoding RLPs with LRR structural domains [99]. The transcription products of *GhlncNAT-ANX2* and *GhlncNAT-RLP7* are long non-coding RNAs whose silencing enhances resistance to *Verticillium* cotton wilt, possibly by promoting *lipoxygenase 1* and *lipoxygenase 2* expression [100]. *GbAt11* (AXMN toxin-induced protein-11) is highly resistant to *Verticillium* wilt, and its overexpression stimulates the expression of disease resistance genes, including *FLS2* and *BAK1,* to increase resistance [101]. GhPUB17 is a U-box ubiquitin ligase E3 that can be inhibited by antifungal protein GhCyP3 to negatively regulate *Verticillium* cotton wilt resistance [102]. *GbANS* is involved in the biosynthesis of anthocyanins in cotton and its silencing significantly reduces anthocyanin production and resistance to *Verticillium* cotton wilt [103]. Similarly, *GbCAD1* is involved in cotton phenol synthesis, and its silencing also reduces resistance to *Verticillium* cotton wilt [104]. 

Taken together, these findings suggest that defense genes induced during pathogen attack provide potent protection to the plant by their respective defense mechanisms (Figure 2). However, more studies are required to understand the mechanisms of action of these genes.

## 3. Mechanism of Human Resistance to *Fusarium* and Its Pathogenic Features

*Fusarium* is a naturally widespread, highly adaptable saprophytic fungus that can survive in a wide range of extreme environments [14]. Some *Fusarium* species are non-pathogenic, while others are extremely harmful. Common pathogenic *Fusarium* species include *F. moniliforme*, *F. solani,* and *FOV* [105]. They infect a wide range of crops (cotton, bananas, aubergines, etc.), cause the worldwide spread of many plant diseases, and infect animals such as fish and shrimps [106]. They may even cause skin, corneal, nail, and systemic damage in humans [107].

### 3.1. Fungal Toxins

Fungi are often found in various types of grain in fields and the grain stored in warehouses. This is accompanied by the production of mycotoxins which affect plant regeneration, cause programmed cell death, and lead to disease in humans and animals following accidental consumption of grain contaminated with mycotoxins or through fungi-contaminated grain coming into contact with wounds or eyes [108].

The gastrointestinal tract is a barrier to harmful substances such as pathogens, toxins, and exogenous antigens. It is a habitat for intestinal flora that synergistically protects the host from fungi and plays a key role in regulating the body’s immune system [109,110]. Mycotoxins that enter the body through food initially interact with the gastrointestinal tract, break through its defenses, disrupt its homeostasis, and alter its histomorphology, nutrient absorption, and barrier function, which results in intestinal dysfunction and local immune compromise and may eventually lead to serious illnesses such as systemic chronic mycotoxicosis [110,111]. Mycotoxins can also affect the susceptibility of animals to infectious diseases, such as bacteria, viruses, and parasites, by affecting the innate and acquired immune systems of healthy animals. Thus, exposure of animals to mycotoxins increases their susceptibility to these infectious diseases [112].

Trichothecenes are mainly produced by *F. graminearum* and include two major toxins: T-2 toxin and deoxynivalenol (DON). Their oral ingestion causes toxicity in animals and humans, results in a variety of symptoms (such as vomiting and diarrhea), and may poison white blood cells and affect the host’s immune system [113,114,115]. The toxicity of T-2 and DON is generally mediated by oxidative stress-mediated DNA damage and apoptosis, although they can also bind to the large subunit of the ribosome and interfere with its peptidyl transferase activity to inhibit protein synthesis [116,117].

Fumonisins are mainly produced by *F. verticillioides*; fumonisin B1 (FB1) is the most abundant type found in nature [118]. Epidemiological studies have shown that esophageal cancer in humans and liver cancer in animals are associated with the ingestion of foods containing fumonisins [119,120]. FB1 has a similar structure to that of sphingolipids and can affect sphingolipid metabolism by interfering with ceramide synthase, leading to increased intracellular sphingomyelin levels that ultimately lead to oxidative stress and apoptosis [121,122]. In addition, FB1 may weaken the intestinal barrier by affecting the synthesis of related proteins and lead to immune dysfunction [123,124].

Zearalenone (ZEA) is a fungal toxin mainly produced in grain by *F. graminearum* and *F. culmorum* which may act synergistically with FBs and DON. It is structurally similar to estrogen; therefore, ZEA competes with natural estrogens for estrogen receptor binding and has a major impact on the reproductive system [125]. In addition, ZEA can downregulate tumor suppressor genes in intestinal cells, which can cause cancer [126].

### 3.2. Fusariosis—Fungal Keratitis

Fusariosis is an infectious disease of the skin, eyes, and internal organs mainly caused by *Fusarium* infections [15]. *Fusarium* is an opportunistic pathogen that does not normally cause disease; however, at times it causes local infections in immunocompetent individuals and invasive infections when the body’s immunity is compromised or when there is trauma. Therefore, neutrophils and macrophages play an important role in the immune defense against *Fusarium*, which infects immunocompromised patients in two main ways: (1) through trauma or foreign bodies that cause infection of the patient’s soft tissues or mucous membranes, resulting in keratitis and nail infection, among others [127]; and (2) through rhinitis and pneumonia caused by inhalation or ingestion of *Fusarium* and its toxins [105]. 

Fungal keratitis is the most common type of fusariosis in normal individuals and it has a significant medical and socioeconomic impact [128,129]. Most patients have an infected corneal stroma due to trauma related to agricultural work, particularly in hot and humid areas that lead to an increase in the growth rate of *Fusarium* and mycotoxin levels [130,131]. The next section presents the pathogenic mechanisms of *Fusarium* and the host defense response using the example of fungal keratitis to help advance the prevention and control of fungal keratitis.

The severity of a fungal infection is determined by the strength of the pathogen and the immunity of the host [129]. The innate immune response against fungal keratitis involves detection of the fungus by macrophages residing in the cornea followed by chemokine secretion by the macrophages, which recruits neutrophils to the infected corneal stroma [132]. The neutrophils act as killer cells and remove the invading fungus. However, insufficient levels of chemokines restrict the number of neutrophils infiltrating the corneal stroma, resulting in a failure to clear the fungus in time for the early stages of infection.

Recognition of fungal invasion by immune cells through the C-type lectin receptor (CLR) and toll-like receptor (TLR) is the first step in the defense response [133]. The CLR and TLR of macrophages recognize intrinsic components of the fungal cell wall, while neutrophils use different oxidative and non-oxidative mechanisms to inhibit mycelial growth and regulate the host immune response to the fungus [134,135]. However, *Fusarium* also expresses a variety of proteins and small molecule virulence factors that inhibit the host’s defense response. For example, conidia secrete the hydrophobic protein RodA, which provides a protective layer for *Fusarium* and prevents recognition by the receptor, thus delaying initiation of the immune response and neutrophil recruitment and allowing the fungus to survive and cause disease [136].

Neutrophils can destroy fungi by producing ROS such as O^2−^, H_2_O_2_, and HClO through the NADPH oxidase (NOX) system [137]. Neutrophils in mice with mutations in the NOX subunit have impaired ability to produce ROS, leading to a marked increase in mycelial growth in the cornea and more severe disease progression [138]. Fungi produce antioxidant factors including thioredoxin and superoxide dismutase (regulated by the Yap1 transcription factor) to counteract neutrophil ROS production. Although most fungal secondary metabolites and catalases have antioxidant activity, they do not enhance fungal resistance to ROS [138]. Protein phosphatase Z (*Ppz*) also plays an important role in oxidative stress tolerance in filamentous fungi [139].

Neutrophils can use various non-oxidative mechanisms to limit microbial growth, such as the production of proteins that compete with microbes for metal ions necessary for growth (a process also known as nutritional immunity) [140]. Iron is essential for thioredoxin production by fungi; neutrophils secrete iron-binding proteins, such as transferrin and lactoferrin, to compete with fungi for iron ions. Zinc is also essential for mycelial growth since it is an important component of many enzymes and TFs, and neutrophils secrete calprotectin, which binds zinc and manganese to compete with fungi [141]. Neutrophils more efficiently kill *Fusarium* species with mutated ion transport carriers, and the growth of these mutant fungi on the cornea itself is inhibited [142]. New mobile, germline-specific chromosomes were identified in *FOV* that can transmit pathogenicity by transfer in plants. These germline-specific chromosomes have numerous genes that encode metal ion transport proteins that may play an important role in the evasion of nutritional immunity by *Fusarium*; these genes are also present in cancer patients with systemic *Fusarium* infections and in those with fungal keratitis [143,144]. 

Macrophages and neutrophils can secrete acidic mammalian chitinase (AMCase) to inhibit fungal growth [145,146]. In vitro experiments showed that neutrophils lacking AMCase are less potent against fungi, while fungi survival is higher in mice lacking AMCase [147]. Neutrophils are an important source of cytokines during infection, producing large amounts of pro-inflammatory and chemotactic cytokines. The expression of IL-1α and IL-1β cytokines is significantly higher in diseased corneas compared to healthy corneas [129,148,149].

Neutrophils are bactericidal in two main ways: direct phagocytosis of pathogens or by the release of a specific structure, namely neutrophil extracellular traps (NETs) consisting of depolymerized chromatin and intracellular granule proteins that cause death to the outside of the cell. This process involves the generation of ROS and the depolymerization of chromatin by neutrophil elastase (NE) to trap and kill pathogenic bacteria [150]. In addition, peptidyl arginine deiminase 4 (PAD4) guanylates histones, which changes the chromatin charge and promotes chromatin depolymerization and nucleus expansion [151]. NETosis is induced by fungi such as *Fusarium*, leading to the release of histones, calprotectin, and other particles. Therefore, NETs kill pathogenic fungi and can also cause damage to their own tissues [152].

## 4. Comparison of Pathogenic Mechanisms against *Fusarium* in Cotton and Humans

Plants and humans possess conserved innate immune systems to respond to microbial invasion (Table 1). They use specialized receptors to sense PAMPs and endogenous DAMPs, which then activates the appropriate defense response. The immune response includes transcriptional reprogramming, production of antimicrobial substances, programmed cell death of infected cells, and the release of soluble factors (including cytokines and phytohormones) that produce a signal from the site of infection to alert the host of the arrival of danger [153].

In vertebrates, membrane-bound immune receptors belonging to the TLR family are located on the cell surface or in endosomal compartments, while the CLR is located on the cell surface. Plant RLKs and RLPs are mainly located on the plasma membrane, with RLPs being structurally similar to TLRs in animal cells [154]. Both plants and vertebrates use membrane PRRs to recognize PAMPs/DAMPs to then trigger activation of transcriptional programs that ultimately produce antimicrobial molecules and enable host adaptive responses to resist pathogen attack.

Activating programmed cell death at the site of pathogen infection is a strategy shared by plants and humans that protects the organism from further invasion by pathogens through directly disrupting the environment in which they live. Cell death also releases intracellular components that can activate or enhance antimicrobial responses and signal alarms to neighboring cells. Apoptosis, pyroptosis, and necroptosis limit pathogen development in humans [155]. Recognition of invading microorganisms by plants also triggers an HR which prevents their spread in healthy tissues by limiting the uptake of plant metabolites by living trophic pathogens [156]. The characteristics of HR-associated cell death are remarkably similar to programmed cell death in humans, and leakage of cell contents can constitute an alarm signal to neighboring cells and prepare them to respond to infection, ensuring a comprehensive and effective defense response [157,158]. In addition, humans and plants secrete chitinase and ROS to inhibit fungal growth [35,77,129,137].

Plants and humans have innate immune memory and immune memory, which enables rapid mobilization of large amounts of resources for the synthesis of disease-resistant proteins that protect them from injury when reinfested by pathogenic bacteria. However, this depends on the plant life cycle, with long-lived perennials exhibiting better immune memory than annuals [159,160].

The biggest difference between animals and plants in the defense against *Fusarium* infestation is that animals have an innate immune system, which is triggered by allergenic proteins such as *Fusarium*’s beta-1,3 glucan and enzymes released during budding. Furthermore, allergen proteins are recognized by phagocytes, neutrophils, and dendritic cells of the host through receptors such as CLR, thereby initiating the allergic process [161,162]. In addition, these receptors activate intracellular signaling pathways that lead to inflammatory responses [163]. Macrophages and neutrophils also inhibit *Fusarium* infection through the presence of interferon-gamma, chemokines, tumor necrosis factor-alpha (TNF-α), granulocyte colony-stimulating factor (G-CSF), granulocyte-macrophage colony-stimulating factor (GM-CSF), and various interleukins (IL-1β, IL-6, IL-15, IL-23) [13,164]. After the innate immune cells are activated, antigen-presenting cells such as macrophages can engulf pathogens and produce antigens, which are then presented to two types of T cells. Helper T cells mainly help B cells to generate antibody-labeled antigens, whereas cytotoxic T cells directly kill pathogen-infected somatic cells [165]. This is centered on neutrophils and is key to an animal’s defense against *Fusarium*. The prognosis for invasive fusariosis in patients with neutropenia or organ transplantation is poor and up to 100% lethal in the absence of recovery of neutrophil function [166]. At the onset of inflammation, killer cells such as neutrophils are attracted by chemotactic substances and rapidly accumulate through blood vessels to the site of disease to surround and destroy the invading *Fusarium* in a single locality and prevent its spread in the body in immunocompetent individuals. 

Unlike animals, plants have not evolved motile immune cells or an adaptive immune system [167]; they also lack a well-developed cellular circulation system similar to that of blood vessels and lymphatic vessels that can produce a direct immune response in the bloodstream. However, every cell in a plant has the ability to sense and resist pathogens [89]. After sensing pathogen signals, plants activate or inhibit TFs through hormone signals such as SA, JA, and ET, thereby affecting the expression of downstream PR genes, the thickening of cell walls, and the production of phytoalexins [168]. Initial animal defenses against pathogens mainly involve the mucous membranes and cuticles. Plant structures also provide physical defense strategies against pathogenic bacteria, such as epicuticular waxes (EWs) and the closure of stomata in many land plants [169,170].

In animals, immune cells such as effector T cells are directed to move flexibly and specifically throughout the body through endogenous cellular factors, such as chemokines or exogenous cellular signals generated by the inflammatory microenvironment, so that resistance is acquired throughout the body [171]. In contrast, plants send a systemic signal at the site of pathogen infestation, such as methyl salicylate (MeSA) which transmits the pathogen signal from the infected site to healthy tissues distal to the plant via intercellular filaments or phloem [172]. This allows the whole plant to acquire resistance to the disease; this process is also known as systemic acquired resistance (SAR) [173]. Systemic resistance in plants is temporarily built up to keep out soon-to-be invading microbes [174]. Conversely, vertebrates have evolved an adaptive immune system with specialized immune cells dedicated to producing a stronger and more specific immune response while ensuring long-term protection of the organism (Figure 3).

Furthermore, in addition to their highly complex adaptive immune system, humans are warm-blooded. Higher body temperature makes the body resistant to most invasive fungal infections, and humans can further raise their body temperature during immune defense; thus, only a few fungi that can grow at or above 37°C can infect humans [175].

## 5. Conclusions and Outlook

There is no cure for *Fusarium* and *Verticillium* cotton wilt caused by *FOV* and *V. dahliae*. Currently, chemical control remains the most efficient way to combat crop diseases, and the application of emerging disease resistance breeding is the most environmentally safe control method. This review summarized the disease resistance and susceptibility genes in various types of cotton and suggested that it is possible to selectively combine or knockdown important disease-related genes to increase disease resistance in cotton [176]. Moreover, these disease-resistant genes can be used in patients undergoing reconstitution of immune function. So far, the treatment of human fusariosis mainly consists of surgical excision of the infected site, systemic antifungal drugs, and reconstitution of immune function in immunocompromised hosts [106,166,177]; if we can find a way to use anti-disease proteins from cotton (such as extracellular enzymes) to assist in the treatment of patients, while avoiding side effects, it may improve the survival rates in critically ill patients and provide a new strategy for the prevention and control of fungal diseases in humans. Knowledge of *Fusarium* infection in humans can also be applied to cotton. Even among agricultural workers, Fusarium infection mostly occurs in the cornea, as the rest of the body is protected by clothing. A major reason for plant susceptibility to *Fusarium* wilt is that *FOV* inhabits the soil, which allows it to invade the plant through its root system. Therefore, the application of techniques such as soilless cultivation will isolate plants from the natural environment of *FOV*, thereby protecting the plant’s vulnerable roots and greatly reducing the chances of *FOV* infestation.

However, the extremely complex pathogenesis of *Fusarium* and *Verticillium* cotton wilt, as well as the large breeding effort, long lead time, development of fungal resistance, and increasing national attention to pesticide residues and food safety issues, has slowed the development of disease resistance breeding and chemical control. Therefore, new ideas are required to prevent fungal diseases. Similar to the way humans and animals are vaccinated against disease, plants can also use non-pathogenic *Fusarium* to effectively control *Fusarium* disease. Some non-pathogenic *Fusarium* directly inhibit the growth, invasion, and colonization of pathogens by competing for nutrients in the soil, infestation sites on the surface of plant roots, and colonization sites within plant tissues [178,179,180]. Other non-pathogenic *Fusarium* can activate the plant’s defense response by colonizing the plant and allowing it to acquire induced systemic resistance, thus indirectly antagonizing the pathogenic *Fusarium* and reducing the damage caused by it [181]. Furthermore, there are viruses among plant pathogenic fungi that can cause fungal virulence decline. These fungal viruses can regulate the expression of the host enzymes and TFs, inhibit the synthesis of the host cytoplasmic membrane, and impair the host’s transport system [182,183,184]; this affects the host’s physiological state and potentially prevents and controls plant fungal diseases. Many fungal viruses that can inhibit *Fusarium* growth and mycotoxin production have been found [185,186,187], and they are expected to be used in the prevention and control of *Fusarium* in the future.

Although *Fusarium* is a vicious pathogen, it can produce many beneficial products. L-tryptophan can induce marine *Fusarium* sp. L1 to produce indole alkaloids with anti-Zika virus activity [188]. The fermentation of *Fusarium* sp. DCJ-A produces cyclic hexapeptide compounds that exhibit cytotoxic activity against five human tumor cells [189]. Xylanase produced by *F. graminearum* FGSG_03624 activates the immune response of plants and enhances their resistance to bacterial and fungal pathogens [190]. In addition, *F. graminearum* Ec220 produces a xylanase with low cellulase content when grown on wheat bran [191], which has good commercial prospects. *FOV* PR-33 produces fusarinolic acid and its isomers, dehydrofusaric acid and fusaric acid, which are resistant to pathogenic bacteria and yeasts; these metabolites are potential antibacterial drugs [192]. Bananas are highly susceptible to *Fusarium* infection [193]; therefore, they could be considered for *Fusarium* cultivation for the production of compounds with industrial applications. This would provide theoretical and technological support in the fields of plant disease resistance and human medicine.

## Figures and Tables

**Figure 1 ijms-23-12217-f001:**
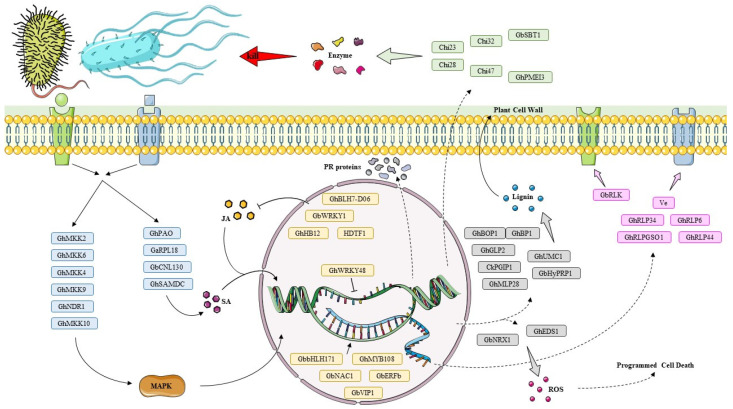
Functional schematic diagram of resistance genes in cotton. Pink boxes represent receptor-like proteins that recognize and bind to extracellular pathogens or damaged plant tissue before then transmitting the signal downstream. Blue boxes represent genes involved in signal transduction, which are important transmitters of signals from the cell surface to the nucleus and ultimately to downstream reaction substrates. Yellow boxes represent transcription factors that interact with RNA polymerase to influence the initiation of the transcription of genes related to disease resistance. Grey boxes are defense-related proteins that both strengthen the defenses of the cell wall, keeping pathogenic bacteria out of the cell, and accumulate ROS levels, thereby activating the cell’s hypersensitivity response, etc. Parts of the figure were drawn using pictures from Servier Medical Art (https://creativecommons.org/licenses/by/3.0/ accessed on 15 July 2022).

**Figure 2 ijms-23-12217-f002:**
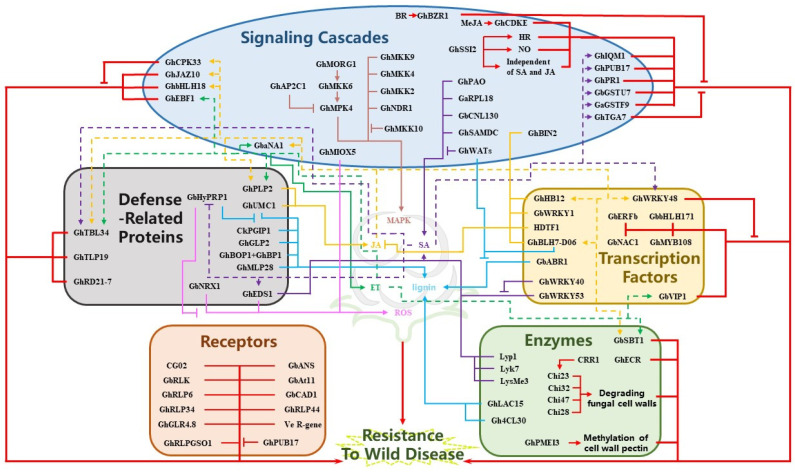
Regulatory network of disease resistance genes in cotton. Solid lines indicate the mechanism by which genes further affect plant disease resistance by acting on JA, SA, ET, lignin, ROS, and MAPK. Dashed lines indicate the mechanisms by which genes are regulated by JA, SA, ET, lignin, ROS, and MAPK to affect plant disease resistance. Blunt ends indicate inhibition. Arrows indicate promotion. Brown, yellow, purple, green, blue, and pink indicate MAPK, JA, SA, ET, lignin, and ROS interactions with these genes, respectively. Red indicates unclear mechanisms.

**Figure 3 ijms-23-12217-f003:**
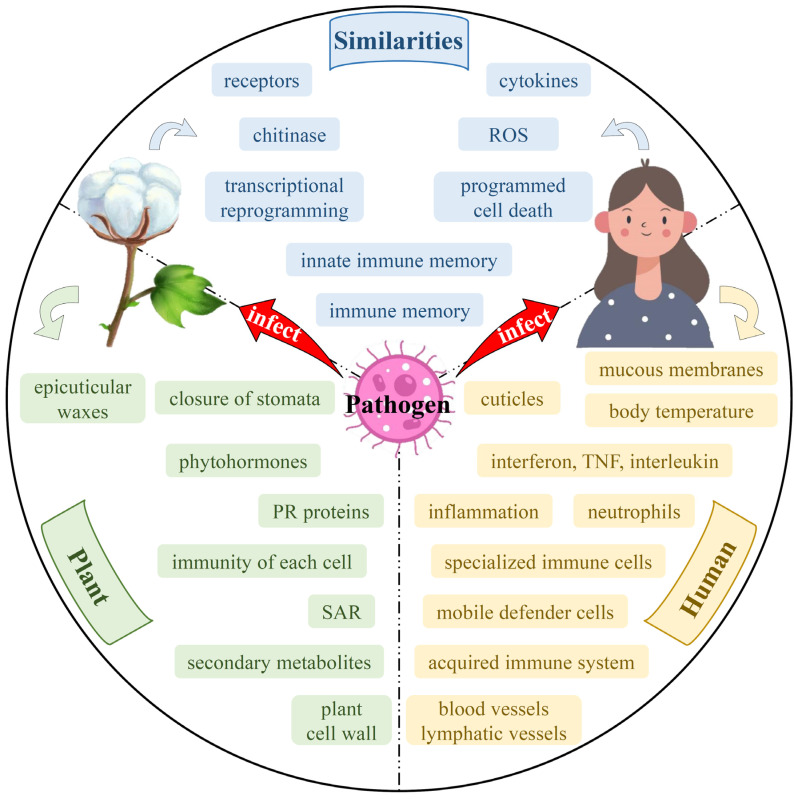
Similarities and differences in the resistance of plants (cotton) and humans to *Fusarium*. Blue represents the similarities between the defense mechanisms of plants and humans; green and yellow represent the respective characteristics of plants and humans. This cover has been designed using images from Freepik.com.

**Table 1 ijms-23-12217-t001:** Comparison between the pathogenic response in humans and cotton.

**Similarities**	**Plant/Humans**	**Receptors**		**Immune Response**			**Natural Strategies**
		Can sense PAMP and DAMP		Transcriptional reprogramming Production of antimicrobial substances Programmed cell death of infected cells Release of soluble factors Secrete chitinase and ROS			Innate immune memory and immune memory
**Differences**		Host Barrier	Whole Body Immunization	Immune Cells	Signaling Pathways	Special Antibacterial Substances	Other Characteristics
	**Plant**	EWs and closure of stomata	SAR	All cells can produce immune functions	Phytohormones	Secondary metabolites	PR proteins Cell wall
	**Humans**	Cuticles and mucous membranes	Movement of immune cells	Specialized immune cells	Interleukin, TNF	Interferon	Innate immune system cells; inflammation; acquired immune system; blood vessels and lymphatic vessels; body temperature

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
