# Peer review of "Defense Mechanisms of Cotton Fusarium and Verticillium Wilt and Comparison of Pathogenic Response in Cotton and Humans"

_ijms, 2022, doi:10.3390/ijms232012217_

Round 1

Reviewer 1 Report

The article by Man et. al. titled with “The defense mechanisms……..and humans”, focused on the general host defense mechanisms of the "cotton" and "human" systems.

The following few points need to address to further improve this manuscript.

 Authors are suggested to modify the title. 

Authors have taken into consideration two broad host systems and have included two pathogens Verticillium along with Fusarium in some sections while in others the same is missing. A focused and specialized analysis is needed throughout the manuscript. 

What does "two pathogens" on the 23rd line of the abstract mean? Verticillium and Fusarium, perhaps? 

In the introduction section, the authors should mention the rationale of this study in more detail. 

It is not entirely apparent how the defensive mechanisms of cotton and humans against the chosen pathogen are similar and different. It is advised that authors critically review the information in section 4 in more depth. Include a table that compares and contrasts the ways that humans and cotton respond to pathogens. 

Current Figure 3 should be improved to make it more interactive for readers. which pathogen? Specify.

Author Response

Response to Reviewer 1 Comments

Dear Reviewers:

Thank you for your comments on our manuscript entitled "The defense mechanisms against pathogenic Fusarium in cotton and humans" (ijms-1905559). Those comments were very helpful for improving our manuscript. We believed we have answered reviewer’ s all comments. The main corrections in the paper and the responds to the reviewer’s comments are as follows:

Comments to the Author

The article by Man et. al. titled with “The defense mechanisms……and humans”, focused on the general host defense mechanisms of the "cotton" and "human" systems.

The following few points need to address to further improve this manuscript.

Comment 1: Authors are suggested to modify the title.

Response 1: Thank you for your suggestion. We have modified the title (L2-L3). The new title is “Defense Mechanisms Against Pathogenic Fungi in Cotton and Humans”.

Comment 2: Authors have taken into consideration two broad host systems and have included two pathogens Verticillium along with Fusarium in some sections while in others the same is missing. A focused and specialized analysis is needed throughout the manuscript.

Response 2: Thank you for your comment. We have replaced the "pathogenic Fusarium" in the title to "Pathogenic Fungi" (L2), and then discussed in the first and third paragraphs of the introduction section respectively why the plant describes two fungi but the human had only one.

In the paragraph 1 of Part 1 (L46-L58), we have described that the main pathogenic fungi that infect plants are Fusarium and Verticillium, both of which are mainly responsible for yield reduction in cotton, so we using these two fungi as examples in the plant section.

In the paragraph 3 of Part 1 (L73-L83), we have added “Humans are warm-blooded, and their innate and adaptive immune system is complex; humans are naturally resistant to most invasive fungal infections. For fungi to successfully infect humans they must: (1) grow at or above human body temperature (37 °C), (2) bypass or penetrate host surface barriers to reach internal tissues, (3) digest and absorb human tissues, and (4) resist against the human immune system. Thus, the number of fungi capable of infecting a healthy human host is very limited [11,12]. However, the pathogenic bacteria Fusarium and Aspergillus have these abilities and can therefore infect humans. Although there are limited data on the role of host defenses against the fungi, fusariosis and aspergillosis, as well as other fungal infections, share many features[13]. Hence, this review discusses Fusarium to illustrate the relationship between pathogenic fungi and humans.” to explain why Verticillium dahliae cannot infect humans, so we have only used Fusarium as example in the human part.

Comment 3: What does "two pathogens" on the 23rd line of the abstract mean? Verticillium and Fusarium, perhaps?

Response 3: Thank you for your comment. The "two pathogens" on the original 23rd line of the abstract should only mean Fusarium, I made a mistake, so we have modified the abstract (L21-L31). The new abstract is “Cotton is an important economic crop. Fusarium and Verticillium are the primary pathogenic fungi that threat both the quality and sustainable production of cotton. As an opportunistic pathogen, Fusarium causes various human diseases, including fungal keratitis, which is the most common in humans. Therefore, it is urgent to study and clarify the resistance mechanism of cotton and humans to Fusarium to mitigate, or eliminate, its harm. Herein, we first discuss the resistance and susceptibility mechanisms of cotton to Fusarium and Verticillium wilts and classify associated genes based on their functions. Then, we outline the characteristics and pathogenicity of Fusarium and describe the multiple roles of human neutrophils in limiting hyphal growth. Finally, we comprehensively compare the similarities and differences between animal and plant resistance to Fusarium and put forward new insights into novel strategies for cotton disease resistance breeding and treatment of Fusarium infection in humans”.

Comment 4: In the introduction section, the authors should mention the rationale of this study in more detail.

Response 4: Thank you for your suggestion. We have overhauled the introduction section. We have modified the paragraph 1 of Part 1 (L46-L58). The new paragraph 1 is “Cotton (Gossypium spp.) is an important cash crop as it yields grain, fiber, and oil [1]. Cotton is the world's most important source of natural fiber, providing approximately 35% of the world's total fiber [2]. Compared with synthetic fiber, cotton, as a renewable resource, as cotton plantations generate a higher number of jobs that involve planting, processing and textile manufacturing. Cotton, has important environmental and social benefits, and cottonseed can be used as animal feed after oil extraction [3]. Cotton is grown in more than 80 countries, of which approximately 30 regard cotton as a major crop [4]. China is the world's largest producer of cotton fiber [5]. However, cotton has been affected by various pests and pathogens, among which Fusarium and Verticillium wilts cause serious global cotton economic losses [6,7], and are considered the main obstacle to sustainable high-quality cotton production in China [8]. Therefore, this review describes the relationship between cotton and pathogenic fungi, using Fusarium and Verticillium as examples”. We provide a more detailed information of cotton and show that Fusarium and Verticillium are the main fungi responsible for its growth and yield.

We have added “Fusarium oxysporum (FOV) is the cause of Fusarium wilt in cotton and is classified into 8 physiological types, of which 3 exist in China [9]. There are two species of Verticillium dahliae that infect cotton, namely deciduous and non-deciduous, based on the symptoms [10]. These pathogens can exist in soil and plant debris in the form of mycelium, chlamydia spores, and microspores and survive in soil for a long time until the beginning of a new infection cycle [9]. In addition, they can survive saprophytically on other crops and weeds [8]. Diseases caused by Fusarium typically manifest following the seedling stage, and are most severe during the budding stage. The onset of Fusarium wilt is characterized by plaques between the main veins, while the rest of the leaves remain green, and chlorotic plaques gradually cause the leaves to become necrotic and fall off the stem [9]. Verticillium wilt occurs before the budding stage, manifesting as yellow mottled or fallen leaves during the boll setting stage, when the disease is also the most serious [10]. However, the determinants that lead to the development of these diseases remain unclear [8]” as a new paragraph 2 (L59-L72). We describe a comprehensive overview of the symptomatic features caused by Fusarium and Verticillium and the similarities and differences between them.

We have added “Humans are warm-blooded, and their innate and adaptive immune system is complex; humans are naturally resistant to most invasive fungal infections. For fungi to successfully infect humans they must: (1) grow at or above human body temperature (37 °C), (2) bypass or penetrate host surface barriers to reach internal tissues, (3) digest and absorb human tissues, and (4) resist against the human immune system. Thus, the number of fungi capable of infecting a healthy human host is very limited [11,12]. However, the pathogenic bacteria Fusarium and Aspergillus have these abilities and can therefore infect humans. Although there are limited data on the role of host defenses against the fungi, fusariosis and aspergillosis, as well as other fungal infections, share many features[13]. Hence, this review discusses Fusarium to illustrate the relationship between pathogenic fungi and humans” in the paragraph 3 (L73-L83) to present the characteristics required for a fungus to infect humans (Fusarium meets those necessary factors), and the rest describes the current situation of the human risk of Fusarium disease.

We have rewritten the paragraph 4 of Part 1 as “In order to eliminate, or significantly mitigate, the harm of cotton Fusarium and Verticillium wilt to improve yield, and at the same time seek a method to treat patients suffering from Fusarium spp. infectious diseases, we review the studies on cotton genes related to resistance and susceptibility to Fusarium and Verticillium wilt in recent years, summarize the molecular mechanisms by which humans resist Fusarium disease, compare resistance strategies in humans and plants, and provide novel insights into the improvements in the resistance of plants and humans against Fusarium.” in the manuscript (L89-L95) in order to make the purpose and meaning of this review clear to the readers.

Comment 5: It is not entirely apparent how the defensive mechanisms of cotton and humans against the chosen pathogen are similar and different. It is advised that authors critically review the information in section 4 in more depth. Include a table that compares and contrasts the ways that humans and cotton respond to pathogens.

Response 5: Thank you for your suggestion. We have revised the section 4 completely. We have modified “adaptive” to “innate” (L592) and added “which is triggered by allergenic proteins such as Fusarium's beta-1,3 glucan and enzymes released during budding. Furthermore, allergen proteins are recognized by phagocytes, neutrophils, and dendritic cells of the host through receptors such as CLR, thereby initiating the allergic process [164,165]. In addition, these receptors activate intracellular signaling pathways that lead to inflammatory responses [166]. Macrophages and neutrophils also inhibit Fusarium infection by interferon-gamma, chemokines, tumor necrosis factor-alpha (TNF-α), granulocyte colony-stimulating factor (G-CSF), granulocyte-macrophage colony-stimulating factor (GM-CSF), and various interleukins (IL-1β, IL-6, IL-15, IL-23) [13,167]. After the innate immune cells are activated, antigen-presenting cells such as macrophages can engulf pathogens and produce antigens, which are then presented to two types of T cells. Helper T cells mainly help B cells to generate antibody-labeled antigens, while cytotoxic T cells directly kill virus-infected somatic cells [168].” in the paragraph 5 of section 4 (L592-L604).

We have added “But every cell in a plant has the ability to sense and resist pathogens [90]. After sensing pathogen signals, plants activate or inhibit TFs through hormone signals such as SA, JA, and ET, thereby affecting the expression of downstream PR genes, the thickening of cell walls, and the production of phytoalexins [171].” in the paragraph 6 of section 4 (L614-L617).

We have added “Furthermore, humans not only have a highly complex adaptive immune system, but are also warm-blooded. Higher body temperature makes the body resistant to most invasive fungal infections, and during immune defense, humans can raise body temperature again, so only a few fungi that can grow at or above 37°C can infect humans [176].” as the paragraph 8 of section 4 (L633-L636).

Part of the original paragraph 4 (“Initial animal defenses against pathogens mainly involve the mucous membranes and cuticles. Plant structures also provide physical defense strategies against pathogenic bacteria, such as epicuticular waxes (EWs) and the closure of stomata in many land plants [160,161].”) was inserted as differences in paragraph 6 (L617-L620), and add the rest (“In addition, humans and plants secrete chitinase and ROS to inhibit fungal growth [37,80,130,138].”) to the end of paragraph 3 (L579-L580).

We have modified Figure 3 accordingly, as seen in Comment 6.

We have added the table that compares the ways that humans and cotton respond to pathogens (Table 1) in the manuscript (L559-L560).

Table 1. A comparison of the ways that humans and cotton respond to pathogens

Similarities

Plant/Humans

Receptors

Immune response

can sense PAMP and DAMP

transcriptional reprogramming, production of antimicrobial substances, programmed cell death of infected cells and release of soluble factors.

secrete chitinase and ROS

have innate immune memory and immune memory

Differences

Host barrier

Whole body immunization

Immune cells

Signaling pathways

Special antibacterial substances

Other characteristics

Plant

EWs and closure of stomata

SAR

all cells can produce immune functions

phytohormones

Secondary metabolites

PR proteins,

Cell wall

Human

Cuiticles and mucous membranes

movement of immune cells

specialized immune cells

Interleukin, TNF

interferon

Innate immune system cells neutrophils and macrophages,

Inflammation, acquired immune system, blood vessels and lymphatic vessels, body temperature

Comment 6: Current Figure 3 should be improved to make it more interactive for readers. which pathogen? Specify.

Response 6: Thank you for your comment. We have improved the Figure 3 in the manuscript (L637), and we have replaced “The regions in the middle are the same or similar, and the left and right regions are features specific to cotton and humans, respectively.” with “The blue area shows the similarities between the defence mechanisms of plants and humans, while the green area on the left and the yellow area on the right show the characteristics of the respective defence mechanisms of plants and humans. This cover has been designed using images from Freepik.com.” in the manuscript (L640-L643).

Figure 3. Similarities and differences in the resistance of plants and humans to Fusarium. Shown are the similarities and differences in the defense responses of cotton and humans after infection with pathogens. The blue area shows the similarities between the defence mechanisms of plants and humans, while the green area on the left and the yellow area on the right show the characteristics of the respective defence mechanisms of plants and humans. This cover has been designed using images from Freepik.com.

Reviewer 2 Report

In section 2.1, the authors summarized a lot of genes related to infection. They are not organized well, so it is a bit confusing to read. For example, the WRKY family proteins are TFs. But some of them are included in the second paragraph of 2.1, while some of them are included in L152-L155. Besides, the relationship of GhWRKY40 and MAPK cascade, or GhNDR1 and MAPK cascade was not indicated clearly. The WAT genes are reviewed separately in L137 and L263. “Transcription factors are divided into two paragraphs by the hormone signaling pathways. I strongly suggest the authors to re-organize the manuscript.

In L56 and L57, “fungus” should be deleted.

The authors not only summary the molecular mechanisms of disease resistant genes, but also include some disease susceptible genes. However, in Introduction and Conclusion, only resistant genes are mentioned.

Title 3. should be revised. Pathogenesis belongs to the pathogen, while resistance belongs to host.

I do not think it is necessary to discuss so much about human immune cells, since they do not exist in plants. There is no comparability.

Author Response

Thank you for your comments on our manuscript entitled "The defense mechanisms against pathogenic Fusarium in cotton and humans" (ijms-1905559). Those comments were very helpful for improving our manuscript. We believed we have answered reviewer’ s all comments. The main corrections in the paper and the responds to the reviewer’s comments are as follows:

Comments and Suggestions for Authors

Comment 1: In section 2.1, the authors summarized a lot of genes related to infection. They are not organized well, so it is a bit confusing to read. For example, the WRKY family proteins are TFs. But some of them are included in the second paragraph of 2.1, while some of them are included in L152-L155. Besides, the relationship of GhWRKY40 and MAPK cascade, or GhNDR1 and MAPK cascade was not indicated clearly. The WAT genes are reviewed separately in L137 and L263. “Transcription factors” are divided into two paragraphs by the “hormone signaling pathways”. I strongly suggest the authors to re-organize the manuscript.

Response 1: Thank you for your suggestion. We replace the first paragraph of section 2.1 with “During the long-term evolution of plants, an immune system directly controlled by a series of cross-linked signal transduction pathways and various plant hormones has been formed to rapidly regulate gene expression, thereby adapting, resisting and tolerant to various biotic stresses. The reaction substrates of these intricate signaling networks are usually transcription factors (TFs). TFs may be regulated via multiple signals and may also regulate the expression of multiple genes related to disease resistance. They also play an important regulatory role in the process of plant stress signal transmission” (L140-L145). And we have deleted the “Plant transcription factors (TFs) are extremely complex in their response to biotic stresses and are involved in complex interactions between different signaling pathways” (L235-L236) in the manuscript. In addition, we have replaced the “The MYB family” with “The MYB (v-myb avian myeloblastosis viral oncogene homolog) transcription factor family” in the manuscript (L237).

We moved “Within this intricate signaling network, the mitogen-activated protein kinase (MAPK) cascade response is the primary pathway for sorting and amplifying external signals into intracellular signals, including three kinases: MAPK kinase kinases (MAPKKKs), MAPK kinases (MKKs) and MAPKs, which are important in response to biotic- and abiotic stresses” to the beginning of paragraph 2 in section 2.1 (L152-L156). And we have added “Other members of the WRKY (pronounced ‘worky’) family are regulated by other hormones” in the manuscript (L163-L164). We also moved “GbWRKY1 is a negative regulator of the JA-mediated defense pathway involved in plant resistance to V. dahliae and mediates the transition from defense to development by activating JAZ1 expression during infection by V. dahliae. GhWRKY48 expression in roots, stems, and leaves is induced by V. dahliae infestation, JA, and SA; its silencing enhances resistance to Verticillium wilt in cotton and may act as a negative regulatory transcription factor to repress the expression of downstream disease resistance genes, thereby affecting cotton resistance to Verticillium wilt.” to the second paragraph of section 2.1 (L164-L170).

We added “GhWRKY40 is a substrate of GhMPK20,” to the second paragraph of section 2.1 (L158) but we did not find a direct relationship between GhMKK2 and GhNDR1 after reviewed the literatures. And we moved “Walls are thin (WAT) genes regulate SA metabolism and signaling by affecting polar transport of growth hormone to further enhance plant resistance to a variety of pathogens. Knockout of GhWATs increases SA accumulation, activates the expression of SA path-way-related genes, and increases lignin accumulation in xylem, each of which promotes plant resistance to Verticillium wilt resistance.” to the seventh paragraph of section 2.2 (L358-L362).

Comment 2: In L56 and L57, “fungus” should be deleted.

Response 2: Thank you for your suggestion. We have removed the “fungus” in the manuscript (L118 and L119).

Comment 3: The authors not only summary the molecular mechanisms of disease resistant genes, but also include some disease susceptible genes. However, in Introduction and Conclusion, only resistant genes are mentioned.

Response 3: Thank you for your comment. We have added “and susceptibility” to the abstract (L26), introduction (L92), and conclusion (L647) sections, respectively.

Comment 4: Title 3. should be revised. Pathogenesis belongs to the pathogen, while resistance belongs to host.

Response 4: Thank you for your suggestion. We have modified title 3 to “Mechanism of human resistance to Fusarium and its pathogenic features” in the manuscript (L426).

Comment 5: I do not think it is necessary to discuss so much about human immune cells, since they do not exist in plants. There is no comparability.

Response 5: Thank you for your suggestion. We wanted to compare the similarities and differences between cotton and human defences against Fusarium and use this as a reference to provide new ideas for both defences against Fusarium, which is why we wrote about how humans fight Fusarium. The human resistance to fungal diseases such as Fusarium is largely dependent on innate immune cells such as neutrophils and the acquired immune system based on them, so we discussed a lot about human immune cells. Although it is true that they do not exist in plants, innate immune cells have many similarities to plant defence strategies, such as the production of ROS and chitinases, and structurally similar receptors that can sense pathogens, and this could also provide some new ideas for the treatment of seriously ill patients whose immune systems have collapsed (just like plants). Thank you for your understanding.

Reviewer 3 Report

Comment 1: Please change the title

Comment 2: Your title about Fusarium, why in the first sentence of abstract about Verticillium wilt?

Comment 3: Rewrite the abstract based on your content

Comment 4:  There is no much information about Cotton in Introduction section.

Comment 5: There is no clear basic information about Fusarium and Verticillium in the Introduction section.

Comment 6: First write general disease mechanism about Fusarium then talk about Molecular mechanisms.

Comment 7: If you want to include information about verticillium please write the differences between both fungal species.

Comment 8: One basic piece of information about the Scientific name, first-time genus and species name enough, but most of the places, again and again, used full genus name, please check throughout the manuscript.

Comment 9: Please include one or two tables

Comment 10: Figure 1 and 2, more informative, but not easy for first time readers

Comment 11; Conclusion and outlook like, review, please make it short (one or two paragraphs)

Comment 12: Work is good, but not that much interesting for readers, please improve

Author Response

Comment 1:Please change your title.

Response 1: Thank you for your comment. We have revised the manuscript, changing the title of “The defense mechanisms against pathogenic Fusarium in cotton and humans” to “Defense Mechanisms Against Pathogenic Fungi in Cotton and Humans” (L2-L3).

Comment 2: Your title about Fusarium, why in the first sentence of abstract about Verticillium wilt?

Response 2: Thank you for your comment. I changed the title's “pathogenic Fusarium” to “pathogenic fungus”, and added a new paragraph to the plant section to say that Fusarium and Verticillium wilts are the two most damaging pathogenic fungi for cotton. At the same time, the abstract has also been revised to explain that Verticillium dahliae is also one of the main culprits of the serious loss of cotton yield, so I use these two fungi as the main objects to expand the elaboration of the plant's disease resistance mechanism.

Comment 3: Rewrite the abstract based on your content.

Response 3: Thank you for your comment. We have replaced the abstract with “Cotton is an important economic crop. Fusarium and Verticillium are the primary pathogenic fungi that threat both the quality and sustainable production of cotton. As an opportunistic pathogen, Fusarium causes various diseases, including fungal keratitis, which is the most common in humans. Therefore, it is urgent to study and clarify the resistance mechanism of cotton and humans to Fusarium to mitigate, or eliminate, its harm. Herein, we first discuss the resistance and susceptibility mechanisms of cotton to Fusarium and Verticillium wilts and classify associated genes based on their functions. Then, we outline the characteristics and pathogenicity of Fusarium and describe the multiple roles of human neutrophils in limiting hyphal growth. Finally, we comprehensively compare the similarities and differences between animal and plant resistance to Fusarium and put forward new insights into novel strategies for cotton disease resistance breeding and treatment of Fusarium infection in humans” (L21-L31).

Comment 4: There is no much information about Cotton in Introduction section.

Response 4: Thank you for your comment. We have added more information about Cotton in the Introduction section. We have replaced the first paragraph of the introduction with “Cotton (Gossypium spp.) is an important cash crop as it yields grain, fiber, and oil. Cotton is the world's most important source of natural fiber, providing approximately 35% of the world's total fiber. Compared with synthetic fiber, cotton, as a renewable resource, as cotton plantations generate a higher number of jobs that involve planting, processing and textile manufacturing. Cotton, has important environmental and social benefits, and cottonseed can be used as animal feed after oil extraction. Cotton is grown in more than 80 countries, of which approximately 30 regard cotton as a major crop. China is the world's largest producer of cotton fiber. However, cotton has been affected by various pests and pathogens, among which Fusarium and Verticillium wilts cause serious global cotton economic losses, and are considered the main obstacle to sustainable high-quality cotton production in China. Therefore, this review describes the relationship between cotton and pathogenic fungi, using Fusarium and Verticillium as examples” in the manuscript (L46-L58).

Comment 5: There is no clear basic information about Fusarium and Verticillium in the Introduction section.

Response 5: Thank you for your comment. We have added clear basic information about Fusarium and Verticillium in the Introduction section. We have added “Fusarium oxysporum (FOV) is the cause of Fusarium wilt in cotton and is classified into 8 physiological types, of which 3 exist in China [9]. There are two species of Verticillium dahliae that infect cotton, namely deciduous and non-deciduous, based on the symptoms [10]. These pathogens can exist in soil and plant debris in the form of mycelium, chlamydia spores, and microspores and survive in soil for a long time until the beginning of a new infection cycle [9]. In addition, they can survive saprophytically on other crops and weeds [8]. Diseases caused by Fusarium typically manifest following the seedling stage, and are most severe during the budding stage. The onset of Fusarium wilt is characterized by plaques between the main veins, while the rest of the leaves remain green, and chlorotic plaques gradually cause the leaves to become necrotic and fall off the stem [9]. Verticillium wilt occurs before the budding stage, manifesting as yellow mottled or fallen leaves during the boll setting stage, when the disease is also the most serious [10]. However, the determinants that lead to the development of these diseases remain unclear [8].” in the manuscript (L59-L72).

Comment 6: First write general disease mechanism about Fusarium then talk about Molecular mechanisms.

Response 6: Thank you for your comment. We have added general disease mechanism about Fusarium in the manuscript (L59-L72), but as it says, the disease mechanism of Fusarium has not been identified clearly.

Comment 7: If you want to include information about Verticillium please write the differences between both fungal species.

Response 7: Thank you for your comment. I have added " Fusarium oxysporum (FOV) is the cause of Fusarium wilt in cotton and is classified into 8 physiological types, of which 3 exist in China [9]. There are two species of Verticillium dahliae that infect cotton, namely deciduous and non-deciduous, based on the symptoms [10]. These pathogens can exist in soil and plant debris in the form of mycelium, chlamydia spores, and microspores and survive in soil for a long time until the beginning of a new infection cycle [9]. In addition, they can survive saprophytically on other crops and weeds [8]. Diseases caused by Fusarium typically manifest following the seedling stage, and are most severe during the budding stage. The onset of Fusarium wilt is characterized by plaques between the main veins, while the rest of the leaves remain green, and chlorotic plaques gradually cause the leaves to become necrotic and fall off the stem [9]. Verticillium wilt occurs before the budding stage, manifesting as yellow mottled or fallen leaves during the boll setting stage, when the disease is also the most serious [10]. However, the determinants that lead to the development of these diseases remain unclear [8]." in the paragraph 2 of Part I (L59-L72) to introduce characteristics of each of the two pathogenic fungi.

In addition, I'd like to explain why I'm writing these two fungi: according to the literature, it is not uncommon for cotton Fusarium and Verticillium wilts to co-occur in China; moreover, in the investigation and analysis of Xinjiang wilted cotton, it was found that 89% of the isolated pathogens were Verticillium wilt and 11% were Fusarium wilt. The main pathogenic fungi that infect plants are Fusarium and Verticillium, both of which are mainly responsible for yield reduction in cotton, so we using these two fungi as examples in the plant section (DOI: 10.1094/PDIS-09-20-2038-RE).

Comment 8: One basic piece of information about the Scientific name, first-time genus and species name enough, but most of the places, again and again, used full genus name, please check throughout the manuscript.

Response 8: Thank you for your suggestion. We have revised the full genus names that appear several times in the manuscript, replacing them with abbreviations. For example, we have replaced “F. oxysporum” with FOV in the manuscript (L59, L119, L157, L159, L306, L331, L369, L393, L429, L525, L642, L703, L705, L708 and L723).

Comment 9: Please include one or two tables.

Response 9: Thank you for your suggestion. We have added the table that compares the ways that humans and cotton respond to pathogens (Table 1) in the manuscript (L556-L557).

Table 1. A comparison of the ways that humans and cotton respond to pathogens

Similarities

Plant/Humans

Receptors

Immune response

can sense PAMP and DAMP

transcriptional reprogramming, production of antimicrobial substances, programmed cell death of infected cells and release of soluble factors.

secrete chitinase and ROS

have innate immune memory and immune memory

Differences

Host barrier

Whole body immunization

Immune cells

Signaling pathways

Special antibacterial substances

Other characteristics

Plant

EWs and closure of stomata

SAR

all cells can produce immune functions

phytohormones

Secondary metabolites

PR proteins,

Cell wall

Human

Cuiticles and mucous membranes

movement of immune cells

specialized immune cells

Interleukin, TNF

interferon

Innate immune system cells neutrophils and macrophages,

Inflammation, acquired immune system, blood vessels and lymphatic vessels, body temperature

Comment 10: Figure 1 and 2, more informative, but not easy for first time readers.

Response 10: Thank you for your comment. Figure 1 vividly describes the operation of cotton disease resistance genes. There are few lines, most of which are pictures. I have selected some representative genes and put them on it, just to show the specific functions of these genes. With the description in Figure 1, I believe that readers can easily understand them. The purpose of Figure 2 is to summarize all the disease resistance genes of cotton. The visibility of the picture is better than that of the table, which can make readers more interested in my article. Figure 2 is indeed a bit complicated, but each line is clearly visible, although it is numerous but not cluttered. It is similar to the tabular summary in other literatures. Thanks for your understanding.

Comment 11: Conclusion and outlook like, review, please make it short (one or two paragraphs).

Response 11: Thank you for your comment.  We have replaced “Conclusion and outlook” with the following new content (L642-L730):

“There is no cure for cotton Fusarium and Verticillium wilts caused by FOV and V. dahliae. Currently, chemical control remains the most efficient way to combat crop diseases, and the application of emerging disease resistance breeding is the most environmentally safe method of control. This review summarizeds the disease resistance and susceptibleness genes in various types of cotton and suggested that it is possible to selectively combine or knockdown important disease resistance genes to combine resistance [177]. In addition to chemical control and breeding for disease resistance, the use of non-pathogenic Fusarium acnes by plants can be effective against Fusarium wilt, along the lines of vaccination of humans and animals for immunity. Some non-pathogenic Fusarium acnes directly inhibit the growth, invasion, and colonization of pathogens by competing for nutrients in the soil, sites of infestation on the surface of plant roots and sites of colonization within plant tissues [178-180]. Other non-pathogenic Fusarium acnes can activate the plant’s defense response by colonizing the plant and allowing the plant to acquire induced systemic resistance, thus indirectly antagonizing the pathogenic Fusarium and reducing the damage caused by it [181]. Furthermore, there are viruses among plant pathogenic fungi that can cause fungal virulence decline. These fungal viruses can regulate the expression of the host enzymes and TFs, inhibit the synthesis of the host cytoplasmic membrane, and impair the host’s transport system [183-185]. This affects the host’s physiological state and potentially prevents and controls plant fungal diseases. It is also interesting that the human defense against epidermis is a good inspiration, even for those working in agriculture, the highest incidence of Fusarium disease is only corneal infection. A major reason why plants are susceptible to Fusarium wilt is that FOV inhabits the soil, which allows it to invade the plant through its root system. Therefore, the application of techniques such as soilless cultivation will help keep plants away from the natural environment of FOV. This strategy protects the plant’s most vulnerable roots and greatly reduces the chances of FOV infestation.

The treatment of human fusariosis mainly consists of surgical excision of the infected site, systemic antifungal drugs, and reconstitution of immune function in immunocompromised hosts [107,169,197]. Patients undergoing reconstitution of immune function may benefit from adjuvant therapy using anti-disease proteins from cotton with no side effects in humans, such as extracellular enzymes. This may improve survival rates in critically ill patients and provide a new strategy for the prevention- and control of fungal diseases in humans. At the same time, Fusarium can produce many beneficial products. L-tryptophan can induce marine Fusarium sp. L1 to produce indole alkaloids with anti-Zika virus activity [198]. The fermentation of Fusarium sp. DCJ-A produces cyclic hexapeptide compounds that exhibit cytotoxic activity against five human tumor cells [199]. Xylanase produced by F. graminearum FGSG_03624 activates the immune response of plants and enhances their resistance to bacterial- and fungal pathogens [200]. In addition, F. graminearum Ec220 produces a xylanase with low cellulase content when grown on wheat bran [201], which has good commercial prospects. FOV PR-33 produces fusarinolic acid and its isomers, dehydrofusaric acid and fusaric acid which are resistant to pathogenic bacteria and yeasts; these metabolites are potential antibacterial drugs [202]. Bananas are highly susceptible to Fusarium infection [203]; hence, these they could be considered for Fusarium cultivation for the production of compounds with industrial applications. This would provide theoretical and technological support in the fields of plant disease resistance and human medicine.”.

Comment 12: Work is good, but not that much interesting for readers, please improve.

Response 12: Thank you for your comment. This article not only reviews the genes for resistance to Fusarium and Verticillium wilts in cotton, it is interesting to compare the way animals and plants are protected against Fusarium wilt. Readers will know this when they see the title and abstract, which is rarely seen in other articles. And at the end of the article, I put forward a lot of novel ways to control Fusarium for cotton and humans, and even we can use Fusarium to produce their rare secondary metabolites for us, I think this is also an interesting place. I'll keep working on it in future writing to be more interesting, thanks for the suggestions.

Round 2

Reviewer 2 Report

Authors have addressed most of my comments. English writing need to be improved. For example, Line521, has been formed should be changed to have been synthesized. Why use Verticillium wilts, but Fusarium wilt? Sometimes use Verticillium wilts, sometimes Verticillium wilt? In Line 75 and 77, oxysporum and dahliae should also be Italic. Line 77, deciduous and non-deciduous should be changed to defoliating and non-defoliating strains. Microspores should be changed to microsclerotia in Line 79. Line 789, the second resistance should be omited. Just some examples, authors should check carefully throughout the manuscript or ask a native speaker for help. 

Author Response

Response to Reviewer 2 Comments

Dear Reviewer:

Thank you for your comments on our manuscript entitled "The defense mechanisms against pathogenic Fusarium in cotton and humans" (ijms-1905559). Those comments were very helpful for improving our manuscript. We believed we have answered the reviewer’s comments. The main corrections in the paper and the responds to the reviewer’s comments are as follows:

Comments and Suggestions for Authors

Comment 1: Authors have addressed most of my comments. English writing needs to be improved. For example, Line521, has been formed should be changed to have been synthesized. Why use Verticillium wilts, but Fusarium wilt? Sometimes use Verticillium wilts, sometimes Verticillium wilt? In Line 75 and 77, oxysporum and dahliae should also be Italic. Line 77, deciduous and non-deciduous should be changed to defoliating and non-defoliating strains. Microspores should be changed to microsclerotia in Line 79. Line 789, the second resistance should be omited. Just some examples, authors should check carefully throughout the manuscript or ask a native speaker for help.

Response 1: Thank you for your suggestion. Because the article is revised, there may be differences in the display of line numbers. Therefore, I have accepted all the revisions, and the new revisions are highlighted in yellow for your review.

We have replaced “has been formed” to “has been developed” (L112-L113), the subject of “has” is “an immune system”. We have modified “oxysporum” and “dahliae” to italic (L46 and L48). We have replaced “deciduous and non-deciduous” to “defoliating and non-defoliating strains” (L48-L49). And we have modified “microspores” to “microsclerotia” (L50). “Verticillium wilts” in the article is a very controversial place, because I had intended to express “Fusarium and Verticillium wilts”, these are two fungi, so I added “s”; and it is also shown in some other articles in the form of “Fusarium and Verticillium wilts”. But after asking some professionals and reviewing some online searches, I've decided to be uniform and leave all the “Fusarium and Verticillium wilt” without the “s”. Finally, we have entrusted editage to polish the manuscript and the changes are highlighted in yellow.

Reviewer 3 Report

Comment 1: Still not satisfied with the title. There are so many pathogenic fungi, only two fungi you worked on. Please modify the title.

Comment 2: Rewrite the conclusion.

Author Response

Response to Reviewer 3 Comments

Dear Reviewer:

Thank you for your comments on our manuscript entitled "The defense mechanisms against pathogenic Fusarium in cotton and humans" (ijms-1905559). Those comments were very helpful for improving our manuscript. We believed we have answered the reviewer’s comments. The main corrections in the paper and the responds to the reviewer’s comments are as follows:

Comments and Suggestions for Authors

Comment 1: Still not satisfied with the title. There are so many pathogenic fungi, only two fungi you worked on. Please modify the title.

Response 1: Thank you for your suggestion. We have modified the title (L2-L3). The new title is “Defense Mechanisms of Cotton Fusarium and Verticillium Wilt and Comparison of Pathogenic Response in Cotton and Human”.

Comment 2: Rewrite the conclusion.

Response 2: Thank you for your comment. We have replaced “Conclusion and outlook” with the following new content (L582-L634):

There is no cure for Fusarium and Verticillium cotton wilt caused by FOV and V. dahliae. Currently, chemical control remains the most efficient way to combat crop diseases, and the application of emerging disease resistance breeding is the most environmentally safe control method. This review summarized the disease resistance and susceptibility genes in various types of cotton and suggested that it is possible to selectively combine or knockdown important disease-related genes to increase disease resistance in cotton [176]. Moreover, these disease-resistant genes can be used in patients undergoing reconstitution of immune function. So far, the treatment of human fusariosis mainly consists of surgical excision of the infected site, systemic antifungal drugs, and reconstitution of immune function in immunocompromised hosts [106,166,177]; if we can find a way to use anti-disease proteins from cotton with no side effects on humans, such as extracellular enzymes, to assist in the treatment of patients, it may improve the survival rates in critically ill patients and provide a new strategy for the prevention and control of fungal diseases in humans. Knowledge of Fusarium infection in humans can also be applied to cotton. Even among agricultural workers, Fusarium infection mostly occurs in the cornea, as the rest of the body is protected by clothing. A major reason for plant susceptibility to Fusarium wilt is that FOV inhabits the soil, which allows it to invade the plant through its root system. Therefore, the application of techniques such as soilless cultivation will isolate plants from the natural environment of FOV, thereby protecting the plant’s vulnerable roots and greatly reducing the chances of FOV infestation.

However, the extremely complex pathogenesis of Fusarium and Verticillium cotton wilt as well as the large breeding effort, long lead time, development of fungal resistance, and increasing national attention to pesticide residues and food safety issues has slowed the development of disease resistance breeding and chemical control. Therefore, new ideas are required to prevent fungal diseases. Similar to the way humans and animals are vaccinated against disease, plants can also use non-pathogenic Fusarium to effectively control Fusarium disease. Some non-pathogenic Fusarium directly inhibit the growth, invasion, and colonization of pathogens by competing for nutrients in the soil, infestation sites on the surface of plant roots, and colonization sites within plant tissues [178-180]. Other non-pathogenic Fusarium can activate the plant’s defense response by colonizing the plant and allowing it to acquire induced systemic resistance; thus, indirectly antagonizing the pathogenic Fusarium, and reducing the damage caused by it [181]. Furthermore, there are viruses among plant pathogenic fungi that can cause fungal virulence decline. These fungal viruses can regulate the expression of the host enzymes and TFs, inhibit the synthesis of the host cytoplasmic membrane, and impair the host’s transport system [182-184]; this affects the host’s physiological state and potentially prevents and controls plant fungal diseases. Many fungal viruses that can inhibit Fusarium growth and mycotoxin production have been found [185-187], and they are expected to be used in the prevention and control of Fusarium in the future.

Although Fusarium is a vicious pathogen, it can produce many beneficial products. L-tryptophan can induce marine Fusarium sp. L1 to produce indole alkaloids with anti-Zika virus activity [188]. The fermentation of Fusarium sp. DCJ-A produces cyclic hexapeptide compounds that exhibit cytotoxic activity against five human tumor cells [189]. Xylanase produced by F. graminearum FGSG_03624 activates the immune response of plants and enhances their resistance to bacterial and fungal pathogens [190]. In addition, F. graminearum Ec220 produces a xylanase with low cellulase content when grown on wheat bran [191], which has good commercial prospects. FOV PR-33 produces fusarinolic acid and its isomers, dehydrofusaric acid and fusaric acid, which are resistant to pathogenic bacteria and yeasts; these metabolites are potential antibacterial drugs [192]. Bananas are highly susceptible to Fusarium infection [193]; hence, they could be considered for Fusarium cultivation for the production of compounds with industrial applications. This would provide theoretical and technological support in the fields of plant disease resistance and human medicine.